# Environmental Causes of Idiopathic Pulmonary Fibrosis

**DOI:** 10.3390/ijms242216481

**Published:** 2023-11-18

**Authors:** Sheiphali Gandhi, Roberto Tonelli, Margaret Murray, Anna Valeria Samarelli, Paolo Spagnolo

**Affiliations:** 1Division of Occupational and Environmental Medicine, University of California San Francisco, San Francisco, CA 94143-0924, USA; sheiphali.gandhi@gmail.com (S.G.); margaretmurray88@gmail.com (M.M.); 2Respiratory Disease Unit, University Hospital of Modena, Department of Medical and Surgical Sciences, University of Modena and Reggio Emilia, 42125 Modena, Italy; roberto.tonelli@me.com (R.T.); annavaleria.samarelli@unimore.it (A.V.S.); 3Laboratory of Cell Therapies and Respiratory Medicine, Department of Medical and Surgical Sciences for Children & Adults, University of Modena and Reggio Emilia, 41124 Modena, Italy; 4Clinical and Experimental Medicine PhD Program, University of Modena and Reggio Emilia, 42121 Modena, Italy; 5Respiratory Disease Unit, Department of Cardiac, Thoracic, Vascular Sciences and Public Health, University of Padova, 35128 Padova, Italy

**Keywords:** idiopathic pulmonary fibrosis, interstitial lung disease, occupational exposures, environmental exposures

## Abstract

Idiopathic pulmonary fibrosis (IPF), the most common and severe of the idiopathic interstitial pneumonias, is a chronic and relentlessly progressive disease, which occurs mostly in middle-aged and elderly males. Although IPF is by definition “idiopathic”, multiple factors have been reported to increase disease risk, aging being the most prominent one. Several occupational and environmental exposures, including metal dust, wood dust and air pollution, as well as various lifestyle variables, including smoking and diet, have also been associated with an increased risk of IPF, probably through interaction with genetic factors. Many of the predisposing factors appear to act also as trigger for acute exacerbations of the disease, which herald a poor prognosis. The more recent literature on inhalation injuries has focused on the first responders in the World Trade Center attacks and military exposure. In this review, we present an overview of the environmental and occupational causes of IPF and its pathogenesis. While our list is not comprehensive, we have selected specific exposures to highlight based on their overall disease burden.

## 1. Introduction

Idiopathic pulmonary fibrosis (IPF) is a chronic and relentlessly progressive condition characterized by scar tissue formation in the lungs and irreversible organ failure. It commonly affects individuals aged 50 and above and is associated with an unfavorable outlook. The occurrence of IPF ranges from three to nine cases per 100,000 individuals annually in North America and Europe [1]. The defining feature of IPF is the presence of a specific radiological or pathological pattern called usual interstitial pneumonia (UIP), although diagnosis requires the exclusion of all known causes of pulmonary fibrosis like connective tissue disease (CTD), drug-induced pneumonitis, radiation fibrosis, or exposure to certain occupational or environmental hazards [2,3].

The factors that trigger the plethora of profibrotic pathways involved in IPF pathogenesis remain unknown, but advancing age, cigarette smoking, viral infection, chronic aspiration of gastric content and genetic factors are all associated with an increased risk of IPF, probably by causing an alveolar epithelial barrier injury [4]. Inflammation may also contribute to the development of fibrosis in IPF, as suggested by animal models wherein inflammation precedes fibrotic response (and suppression of the inflammation attenuates fibrosis); however, animal models of pulmonary fibrosis only partially recapitulate the complexity of human IPF [5]. In addition, inflammatory changes are generally minimal in IPF [6], and anti-inflammatory therapy with systemic glucocorticoids does not alter the natural history of IPF (and may be associated with poor outcomes) [7].

Most research in this field has primarily concentrated on treatments and the influence of specific comorbidities. Conversely, less emphasis has been placed on investigating the consequences of potentially preventable and avoidable environmental and occupational hazards as potential causes of IPF. There are critical medical and policy needs to address this knowledge gap by clarifying the associations between preventable risk factors of occupational and environmental exposures to inhalational hazards and IPF.

In this review, we present an overview of the environmental and occupational causes of IPF and its pathogenesis. While our list is not comprehensive, we have selected specific exposures to highlight based on their overall disease burden.

## 2. Pathogenesis and Environmental Exposure

### 2.1. Interplay Genetics

Genetically, IPF is a complex disorder inherited in an autosomal dominant fashion with reduced penetrance; however, genetic factors are likely to contribute to both development and progression of the disease [8].

Recently, large Genome-Wide Association Studies (GWAS) have identified several genetic *loci* associated with an increased risk of IPF. These *loci* map within genes potentially involved in disease pathogenesis such as modulation of transforming growth factor (TGF)-β signaling, telomere maintenance, cell adhesion, and host defense. Notably, the genetic variant that confers the higher risk for the onset and development of the disease is located within the promoter region of the *MUC5B* gene (rs35705950). This single nucleotide polymorphism represents to date the strongest and most validated genetic risk factor for both familial and sporadic IPF [9,10]. MUC5B plays an important role in the muco-ciliary clearance that is essential for removing inhaled debris and pathogens in lung homeostasis [11,12]. In this context, carriers of the mutant (T) *MUC5B* allele who are exposed to environmental triggers and air pollutants may display an impaired epithelial repair response to injury [13]. Both common (i.e., polymorphisms) and rare (i.e., mutations) variants have been associated with an increased risk of IPF. Associations with common variants implicate in disease pathogenesis genes involved in mucin production (i.e., *MUC5B*, *MUC2*), cell–cell adhesion (i.e., *DSP*, *DPP9*), cytokines and growth factors (i.e., *IL1RN*, *IL8*, *IL4*, *TGF-β1*), telomere maintenance (i.e., *TERT*, *OBFC1*), innate and adaptive immune response (i.e., toll-like receptor signaling, *TOLLIP*, *TLR3*), and cell cycle regulation (i.e., *KIF15*, *MAD1L1*, *CDKN1A*, *TP53*) [14]. Interestingly, the mutant *MUC5B* allele has also been associated with improved survival. Recently, Van der Vis and colleagues looked at the association between *MUC5B* rs35705950 and survival in a population of European IPF patients and found that carriage of the rs35705950 T allele confers a survival advantage of 16 months in individuals aged over 56 years [15]. Rare variants are identified in about 30% of patients with familial IPF and mainly encode proteins involved in either telomere maintenance such as *TERT*, *TERC* (telomerase RNA component), *TINF2* (*TERF1* interacting nuclear factor 2), *DKC1* (dyskerin), *RTEL1*, *PARN* (poly(A)-specific ribonuclease), *NAF1* (nuclear assembly factor 1 ribonucleoprotein) [16,17] or surfactant production such as *SFPTA1*, *SFPTA2*, *SFPTC* (surfactant protein A1, A2, C), and *ABCA3* (ATP binding cassette subfamily A member 3) [18,19,20]. Recently, Allen and colleagues identified a genetic variant associated with disease progression. Specifically, variant rs115982800 located in an antisense RNA gene for protein kinase N2 (*PKN2*) showed a genome-wide significant association with decline in forced vital capacity (FVC) [21]. PKN2 is a Rho and Rac effector protein and may represent a potential novel therapeutic target in IPF. Ma and co-workers used data from 407,615 participants from the UK Biobank study to investigate the relationships among lifestyle, genetic susceptibility, and risk of IPF [22]. They considered various lifestyle variables together, the most important being smoking, diet, and physical activity, and found that patients with an unfavorable lifestyle and high genetic risk displayed the highest risk of IPF (HR, 7.796; 95% CI, 5.482–11.086) compared with those with a favorable lifestyle and low genetic risk. These data suggest a significant synergic effect between genetic risk and lifestyle on IPF pathogenesis, with 32.7% of IPF risk being caused by the interaction between unfavorable lifestyle and high genetic risk [22].

### 2.2. Interplay Aging and Oxidative Stress

Among multiple factors that have been reported to increase the risk of developing IPF, aging is the most prominent one (Figure 1). Aging is a biological process that involves genomic instability, telomere and mitochondrial dysfunction, epigenetic alterations, cellular senescence, and stem cell depletion [23,24]. In particular, cellular senescence involves cell-cycle arrest and the release of inflammatory cytokines with endocrine, paracrine, and autocrine activities [25]. Environmental factors such as cigarette smoke can lead to cell senescence, thus inducing stable cell-cycle arrest. In addition, aging affects both innate and adaptive immunity, impairing cellular defense against pathogens and environmental insults such as cigarette smoke [26]. Accumulating evidence suggests that IPF results from an aberrant reparative response to recurrent alveolar epithelial injury in genetically susceptible individuals; yet, the mechanisms that lead to cell and tissue senescence remain poorly understood.

Homeostasis of the alveolar epithelium involves the proliferation of type II alveolar epithelial cells (AECII), which play a number of protective roles, including the secretion of pulmonary surfactant [27]. However, several types of injury either genetically determined or induced by environmental factors can affect the ability of AECII to produce surfactant, thus impairing epithelial maintenance. Senescence and abnormal functioning of AECII might contribute to progressive epithelial damage. Recently, Yao and colleagues profiled IPF lung tissue using single-cell RNA sequencing to assess the contribution of epithelial cell senescence to the progression of IPF [28]. They confirmed that AECII isolated from IPF lung tissue exhibit transcriptomic features of cellular senescence. In addition, they demonstrated that senescence rather than loss of AECII promotes progressive fibrosis and that either genetic or pharmacologic interventions targeting p53 activation or eliminating senescent cells may block fibrogenesis [28]. Senescent AECII highly express PDGF, TNF, endothelin-1, CTGF, osteopontin, CXCL12, and plasminogen activator inhibitor 1 (PAI-1), which trigger activation and proliferation of both lung fibroblasts and myofibroblasts [29]. In IPF lung, TGF-β1 increases PAI-1 expression both in AECII, thus inducing cell senescence, and in alveolar macrophages, culminating in Smad3 phosphorylation [30]. In line with the hypothesis that IPF is an age-related disease, various cell types from IPF lung show aging-associated features, such as oxidative stress, that involve the generation of ROS from different sources, including mitochondrial electron transport chain (ETC), endoplasmic reticulum and peroxisomes of AECs, fibroblasts, immune cells, endothelial cells, and vascular smooth muscle cells in the lung niche [31]. Furthermore, in IPF lung, there is evidence of impaired cell detoxification from ROS through small molecule antioxidants (glutathione) or specific enzymes. Finally, oxidative stress might contribute to macrophage polarization, which is a dynamic process of injury repair wherein M1-like cells play a role in the initial phase of tissue injury, while M2-like cells take part in the reparative phase [32]. Similar to AECs, lung fibroblasts, which play a prominent role in the fibrotic process of IPF, might also be affected by cellular senescence [33]. Indeed, compared with age-matched controls, primary lung fibroblasts isolated from lung tissue of patients with IPF display more features of senescence, including telomere shortening, abnormal activation, metabolic reprogramming, mitochondrial dysfunction, insufficient autophagy, and senescence-associated secretory phenotypes (SASP) [34]. In addition, IPF lung fibroblasts display higher expression of β-galactosidase, p16, p21, p53 and proinflammatory cytokines (TNF-α, TGF-β, IL-1β, IL-6, IL-8, IL-10, IL-18), chemokines (CXCL1, MCP-1), growth factors (FGF, CTGF, GM-CSF, M-CSF, PDGF), and matrix metalloproteinases (MMP-2, MMP-3, MMP-9, MMP-10, MMP-12) compared with age-matched controls [35,36]. However, the molecular mechanisms linking aging to increased susceptibility to epithelial damage (induced by, among other factors, cigarette smoke and air pollutants) and how they contribute to the development and progression of IPF remain unclear [37].

### 2.3. Interplay of Health Disparities/Total Health

Health equity is the state in which everyone has a fair and just opportunity to attain their highest level of health. Currently, the burdens of disease and poor health and the benefits of well-being and good health are inequitably distributed globally due to social, economic, and environmental factors. Racial and ethnic minority populations are at the highest risk of morbidity and mortality from health and socioeconomic disparities. For instance, compared with White individuals, Black patients have higher rates of pulmonary involvement with autoimmune disease and may have differential survival of ILD [38,39]. In addition, racial and ethnic minority populations face poor enrolment in ILD registries and clinical trials, which limits our understanding of the relationship between health disparities and racial and ethnic differences in outcomes among patients with pulmonary fibrosis. Recently, Adegunsoye and colleagues used data from prospective clinical registries to compare ages at pulmonary fibrosis-related outcomes and the heterogeneity in survival patterns among Black (*n* = 488), Hispanic (*n* = 319), and White patients (*n* = 3985) [40]. Black patients were significantly younger than Hispanic and White patients at the time of first hospitalization, lung transplant, and death, and also had a higher number of hospitalizations, suggesting that racial and ethnic minority populations tend to experience disparities in pulmonary fibrosis-related outcomes. Environmental and socioeconomic factors may also influence outcomes in patients with IPF. Avitzur and co-workers used the CalEnviroScreen 3.0 (CES) (Sacramento, CA, USA), a tool that combines population, environmental, and pollution vulnerability to quantify environmental burden in 603 patients with IPF identified from a longitudinal database at University of California, San Francisco [41]. A higher CES score, which indicates greater disadvantage, was associated with a lower baseline FVC (*p* = 0.006) and diffusing capacity of the lung for carbon monoxide (*p* < 0.001). In addition, patients in the highest population vulnerability quartile were less likely to be on antifibrotic therapy (OR = 0.33; *p* = 0.001) at time of enrolment, compared with those in the lowest quartile. Overall, these data suggest that socio-environmental disadvantage has meaningful impacts on patients with IPF.

### 2.4. Gender Role in IPF Related to Environmental Exposure

IPF has a clear male predominance, while women are more likely to be affected by CTD-associated ILD [42]. However, how and to what extent sex influences the development and progression of IPF remains unclear. Potential mechanisms may be related to, among other factors, sex hormones, X-linked inheritance as in the short telomere syndrome dyskeratosis congenita wherein males are three times more likely to be affected than females [43], occupational exposures due to different working conditions, smoking habits, and exposure to pollutants and pathogens [44,45]. Sesé and colleagues explored gender differences in 236 patients enrolled in the French multi-centre prospective IPF cohort (COhorte FIbrose, COFI) [46]. They observed that, compared to men, women (*n* = 51, 22%) were significantly less exposed to tobacco smoke and to occupational exposure. Baseline FVC % of the predicted value was also significantly higher in women compared to men. Furthermore, honeycombing and emphysema were significantly less common in women, and fewer women were transplanted compared to men, although median survival did not differ between genders. These findings suggest that women have less advanced disease at diagnosis, possibly due to milder exposure history compared to men. Disease progression and overall survival are comparable regardless of gender, but women seem to have less access to lung transplantation.

## 3. Occupational Exposures

Despite being a diagnosis of exclusion, the risk of developing IPF increases in those occupationally exposed to vapors, dusts, gases, and fumes (VGDF), as demonstrated by multiple retrospective case–control studies of IPF registries. Major meta-analyses of the occupational burden of IPF calculated an odds ratio (OR) of 2.0 [confidence interval (CI) 1.2–3.2] with a population attributable fraction of 26% (CI 10–41%) [47], meaning that approximately one in four cases of IPF can be attributed to occupational exposures to VGDF.

Organic dust, notably agricultural and wood dust, has been found to be associated with an increased risk of IPF based on multiple meta-analyses. In several recent systematic reviews and meta-analyses, there has been a significant risk of IPF found for agricultural dust (OR 1.6–2.35) and wood dust (OR = 1.3–2.0) [47,48,49,50]. The exact mechanism by which organic dust leads to lung fibrosis is not fully understood. However, it is important to acknowledge the possibility of undetected late-stage fibrotic hypersensitivity pneumonitis (HP), which, similar to IPF, can present with a UIP pattern of fibrosis. Therefore, even experienced physicians can sometimes struggle to differentiate between fibrotic HP and IPF, particularly if patients have negative precipitins (antibodies) or have already been removed from the source of exposure during clinical evaluation. A small study conducted by Morell et al. revealed that out of 46 patients initially diagnosed with IPF, 20 were later reclassified as having HP after additional information regarding exposure to animal antigens was obtained [51]. It is also essential to consider the potential role of organic dust as a fibrogenic agent in causing pulmonary fibrosis, a process that has not yet been fully elucidated.

Studies utilizing mineralogical analysis techniques have found metal dusts (e.g., nickel, iron, and aluminum) as well as silica in the lymph node and lung tissue of patients with pulmonary fibrosis [52,53]. In addition, recent meta-analyses calculated an OR of 1.4–2.0 for metal dust [47,48,49,50] and an OR of 1.7 for silica dust [47], although clear causation has not been established. Hypothetically, the association between lung fibrosis and metal dust exposure can be explained by the transition of alveolar cells to mesenchymal cells caused by reactive oxygen species (ROS), while other metals can induce ROS [54,55]. These ROS can induce a fibrotic response that radiographically can appear as UIP. Silica dust exposure is associated with profibrotic response through similar means of induction of free radicals leading to inflammation and fibrosis [56]. Notably, silica particles found in biopsies of IPF patients have also been associated with a more rapid decline of pulmonary function [57]. A recent study demonstrated that 16% to 20% of silica-exposed Korean workers, who had received compensation for pneumoconiosis, had radiographic findings consistent with UIP [58]. However, the possible co-exposure of asbestos in the metalworking and construction industries, especially among those working in steel mills in the 20th century, cannot be discounted [59]. Table 1 provides a non-comprehensive list of known occupational etiologies of ILD.

### Molecular and Cellular Mechanisms of Pulmonary Fibrosis—The Paradigm of Pneumoconiosis

Asbestos is a naturally occurring group of mineral silicate fibers that cause pulmonary and pleural fibrosis, lung cancer, and mesothelioma thorough mechanisms that are incompletely understood. Various animal models have been developed to understand the mechanisms underlying asbestos-induced lung damage. These models have used either asbestos inhalation or intratracheal instillation [60]. When exposed to asbestos, alveolar macrophages become activated and phagocytize asbestos fibers. This results in an increased production of ROS and reactive nitrogen species (RNS), and upregulation of inducible nitric oxide synthase (iNOS), which contribute to lung injury by causing DNA damage and apoptosis, and by inducing protein oxidation, lipid peroxidation, and activation of transcription factors, such as nuclear factor (NF)-κB and activator protein (AP)-1 in macrophages, mesothelial cells, and lung epithelial cells [60,61,62]. In turn, this leads to upregulation of a plethora of inflammatory cytokines (e.g., TNFα, IL-1α, IL-1β, IL-6, and IL-12), and chemokines (e.g., CCL2, CCL4, and CCL5) [63,64]. Long-term exposure to asbestos causes chronic inflammation and increased profibrotic cytokines production leading to accumulation of extracellular matrix (ECM) and pulmonary fibrosis. Macrophages and epithelial cells contribute to ECM production by releasing fibrogenic growth factors, such as platelet-derived growth factor (PDGF) and TGF-β. Theories of disease pathogenesis following asbestos exposure have focused on the participation of iron. Indeed, iron-mediated production of ROS contributes significantly to acute and chronic lung remodeling by releasing a variety of pro-inflammatory mediators, and treatment with an iron chelator or oxide radical scavenger attenuates the fibrogenic effects of asbestos in mice [61,65].

Silica, one of the earliest recognized causes of lung disease, is an amorphous crystalline material found in several types of rock (e.g., granite and quartz). Unlike asbestos, silica-related lung fibrosis appears in the form of nodules, which display a characteristic layered or spiral appearance. Specifically, the nodules have a hyaline centre composed of concentrically arranged collagen fibers, whereas the outer zone contains macrophages, lymphocytes and loosely formed collagen, and is the site of active inflammation and enlargement. Exposure of mice to silica dust through aerosol inhalation or intratracheal instillation leads to the development of silicotic nodules reminiscent of the pulmonary lesions observed in humans following occupational exposure [66]. The persistence of silica within phagocytic cells, mainly due to the sticky nature of the dust, results in a chronic inflammatory response characterized by the accumulation of macrophages, neutrophils, and lymphocytes that release proteolytic enzymes and oxidants, thus contributing to DNA damage and lung injury [67]. Macrophages and neutrophils also release a multitude of cytokines (e.g., TNF-α, IL-1, IL-6, IL-17A), chemokines (CCL2, CCL3, CCL4, CCL17), and growth factors (TGF-β, PDGF, connective tissue growth factor [CTGF]) that amplify the inflammatory response, leading to alveolitis and fibroblast activation [67,68]. Interestingly, however, fibrogenesis may occur independently of inflammation and innate immune response. Re and co-workers determined lung response to silica in knockout mice for myeloid differentiation primary response gene 88 (MyD88), which links members of the toll-like receptor (TLR), nucleotide-binding oligomerization domain receptor (NLR), and interleukin-1 receptor (IL-1R) superfamily to the downstream activation of NF-kappa B and mitogen-activated protein kinases [69]. They showed that MyD88 is crucial for the development of inflammation and granulomas following silica exposure, whereas pulmonary fibrosis in MyD88^−/−^ mice was associated with the accumulation of profibrotic regulatory T lymphocytes, IL-17-producing Th17 lymphocytes, and profibrotic cytokines such as TGF-β, IL-10, and PDGF-B) [68]. The mechanisms underlying the development of silicosis seem to vary between rodent species. Indeed, while in mice, silica-induced fibrosis is associated with significant increases in fibrogenic cytokines including IL-4, IL-13, and IL-10, in rats, silicosis results from chronic inflammation associated with overexpression of TNF-α and IL-1, which are both proinflammatory and profibrotic [60,67].

## 4. Environmental Exposure and Acute Exacerbations

In the complex interplay between environmental factors and IPF pathogenesis, the relationship between environmental triggers and acute exacerbations of the disease (AE-IPF), unpredictable events with poor outcomes, is even more puzzling. AE-IPF are more common in the winter months, suggesting that seasonal factors (i.e., air pollution alongside infectious triggers) may contribute to their onset and development [70]. In this regard, it has been hypothesized that air pollution may act as a repetitive stimulus capable of interfering with telomere shortening and oxidative stress [71]. These mechanisms might amplify lung injury and trigger—or at least enhance—the inflammatory pathways leading to AE-IPF [72,73]. The first clinical evidence linking air pollution to the risk of AE-IPF was reported by Johannson et al. They analyzed retrospectively a cohort of 436 Korean patients with IPF and found that higher exposures to O_3_ and NO_2_ increased significantly the risk of AE-IPF [74]. Sesè and co-workers confirmed short-term exposure to increased O_3_ levels as a risk factor for AE-IPF in a prospective cohort of 192 French IPF patients [75]. Moreover, Yoon et al. demonstrated that increased long-term exposure to NO_2_ increases the risk of mortality, especially in elderly male patients with IPF [76]. The role of PM_10_ and PM_2.5_ in the occurrence of AE-IPF was initially suggested by Johannson et al. [74] and Sesè and al. [75], although a significant association was not found, probably due to exposure misclassification. Conversely, a positive relationship between short-term PM_2.5_ exposure and AE-IPF was reported in a case–control study of 152 Japanese patients with surgically confirmed IPF, in whom an increase of 10 μg/m^3^ in PM_2.5_ amplified the risk of AE of the disease by approximately 2.5-fold [77]. Conversely, no significant relationship was observed between incidence of AE-IPF and exposure to NO_2_ and O_3_, suggesting a potential pathogenetic role of genetic predisposition and ethnic origin [77]. In a recent study, Tomos and co-workers reported significant associations between long-term exposure to increased levels of NO_2_, PM_2.5_, and PM_10_ and the risk of AE-IPF, independent of age, lung function impairment, anti-fibrotic treatment, and smoking status [71]. Further, they found that long-term exposure to O_3_ or PMs is associated with changes in the peripheral blood levels of IL-4, IL-13, and osteopontin, thus providing evidence for a biological link between long-term air pollution exposure and AE-IPF [71]. The documented role of PMs in favoring AE-IPF rises concerns on the potential deleterious effect of climate change, which has been associated with increased incidence of desert dust PM_10_ storms in several areas of the world, including southern Europe, causing a meaningful increase in exacerbations of other chronic respiratory diseases [78].

Although cigarette smoking (both direct and second hand) is one of the most recognized risk factors for the development of IPF [79,80], its role as a trigger for AE is not definitely established. While analysis of the INPULSIS dataset showed that current or former smokers were significantly associated with an increased risk for AE-IPF [70], other observations reported never smoking as a risk factor [81,82]. Further, in IPF, a survival difference favoring current smokers was initially described as a “healthy smoker” effect [83], but this was likely related to lead-time bias from an earlier diagnosis rather than a true biologic mechanism. Indeed, in a more recent study investigating disparities in severity-adjusted survival rates among active and former smokers with IPF, active smokers’ minimal survival benefit faded upon adding age to the model [84]. The role of smoking as a contributing factor to the risk of AE-IPF and its interplay with genetic predisposition and ethnicity need to be clarified by means of adequately designed epidemiological studies.

## 5. Conflict-Related Disasters/Inhalation Exposures

The recent literature on inhalational injuries from conflict-related disasters has focused on the first responders in the World Trade Center (WTC) attacks. These first responders, including police, firefighters, and emergency medical workers, were at the highest risk for potential injuries, as they were exposed to the highest concentration of toxins. High dust exposure following the WTC attacks has been associated with numerous acute and chronic respiratory diseases and symptoms, including IPF. The collapse of the WTC towers released aerosolized contaminants, including heavy metals, silica, asbestos fibers, wood dust, polyaromatic hydrocarbons (PAH), and polychlorinated biphenyls from the jet fuel, burned plastics, cellulose, and other materials [85]. Exposure to many of these inorganic and organic materials, specifically silica, asbestos fibers, heavy metals, and wood dust, have been associated with an increased risk of diffuse interstitial changes, such as those seen in IPF [86,87,88]. The inflammatory response to inhalation of causative chemicals following 9/11 has been proposed as a potential mechanism for the association between WTC dust exposure and IPF. Within the WTC Registry, several cases of IPF were reported less than ten years following September 11, 2001, and subsequently underwent mineralogical analyses of the available tissues, indicating the presence of toxins, including metals, silicas, carbon nanotubes, and chrysotile asbestos [55,89,90]. Later studies indicated that the initial exposure to airborne contaminants unique to WTC makes first responders susceptible to IPF [91]. Additionally, an exposure–response relationship exists between WTC dust exposure and IPF [91].

Outside of the WTC disaster, most of the literature on inhalational injuries from natural disasters has focused on the wildland fires in Australia and Canada and acute inhalational exposures or obstructive lung diseases, rather than ILD. One US study showed that firefighters have a higher risk of ILD and IPF compared to the working population [92], but there is a major research gap to fully elucidate this relationship. Veterans are another occupational cohort with an elevated prevalence of IPF due to military exposure such as in Southeast and Southwest Asia [93]. Veterans in Iraq and Afghanistan have been exposed to benzene, PAH, furans [94], and heavy metals from burn pits [95,96]. These hazards are thought to induce oxidative stress and epigenetic changes, leading to inflammation and fibrosis of the lung parenchyma [97].

A recent study on US veterans showed that the incidence and prevalence of IPF in US veterans was higher than previously thought [93,98]. Additionally, Vietnam veterans exposed to Agent Orange are at higher risk of IPF through unclear mechanisms [99]. There is significant need to fill these research gaps in future investigation.

## 6. Prevention Strategies

While a growing body of evidence strengthens the role of environmental exposures in the pathogenesis and progression of IPF, the implications for the daily clinical management of these patients remain uncertain [100,101]. Yet, if the parenchymal fibrotic changes may be difficult to revert, removing stimuli of cyclic lung injury could contribute to preventing further damage, thus representing a rather cost-effective treatment strategy [102]. In this line, smoking cessation and infection prevention (i.e., influenza and pneumococcal vaccination) seem the most immediate preventative interventions to limit disease progression and improve respiratory function and quality of life [4,100,102,103,104]. Bellou et al., have recently found that active and second-hand cigarette smoking work synergistically in increasing the risk of IPF, and the intensity of tobacco exposure presents a dose–response association with the disease [79]. Hence, from a public health perspective, the strategies aimed at reducing the prevalence of active and second-hand smoking should be reinforced and strengthened in the vulnerable subset of IPF patients. To our knowledge, there are no available data regarding the effect of electronic cigarettes in IPF patients. However, given that vaping has been associated with a wide variety of acute and subacute lung injuries (including lipoid pneumonia, acute respiratory distress syndrome, and diffuse alveolar hemorrhage) [105], its use should be discouraged among patients with chronic respiratory conditions and close monitoring must be granted.

It is still not known if wearing a face mask that filters out air particulates may affect clinical outcome of IPF patients. Moreover, to date, no data are available on the use of home-based air filters or other tools that may mitigate exposure to PM in patients with chronic fibrotic lung disease. In addition, the use of N95 or KF94 masks can interfere with the breathing of patients with advanced respiratory diseases and low pulmonary function, thus leading to uncomfortable respiration, an unsafe breathing pattern, and increased risk of respiratory failure [106]. Properly designed trials aimed at assessing the impact of such preventive measures and tools on clinical outcomes of IPF patients (i.e., acute exacerbation rate, decline in respiratory function, physical performance) are thus warranted. Meanwhile, it seems cautious for IPF patients to stick to regional recommendations to mitigate the effect of air pollution exposure and consider themselves as a “fragile respiratory phenotype” at a higher risk to their inhaled environment.

Understanding the interplay between genetic predisposition and environmental factors could contribute to determining whether there is susceptibility to certain exposures, thus promoting individualized protection approaches. To this purpose, studies should firstly focus on the development of specific and detailed questionnaires designed according to local customs and lifestyle, with the aim to isolate potentially relevant causes amongst the chronic and mixed exposures of a lifetime. Additionally, the identification of specific biomarkers of exposure may ideally contribute to the comprehensive assessment of environmental and professional history. In addition, investigations regarding the effect of environmental exposures on IPF comorbidities (i.e., risk of cancer development) are needed in order to understand the role of preventable traits on the major causes of death in this fragile population. Finally, involving trained professionals (i.e., hygienists) in the objective assessment of exposure within different occupational environments may help to recognize and quantify the extent within the natural course of the disease.

Despite all the difficulties in identifying and counteracting the harmful exposures interacting with the pathogenesis of IPF, we feel that a greater awareness on the potential effect of the environment on clinical outcomes in this subset of patients should be encouraged. Indeed, avoiding or mitigating the preventable environmental exposures could represent an adjunct and cost-effective “treatment strategy” with the aim to impact on quality of life and even prognosis.

## 7. Conclusions

A multitude of occupational/environmental exposures, including, among others, respirable dust exposure in mining, construction, and manufacturing, has been associated with an increased risk of IPF (Figure 2), particularly among racial and ethnic minority populations and in low–middle income countries wherein safety standards may be suboptimal. This highlights the importance of taking a thorough exposure history from all patients in whom IPF is suspected, since the reduction of such exposures in early disease stages may potentially improve outcomes. Air pollution as well as various lifestyle variables, including smoking and diet may also increase the risk of IPF, probably through interaction with genetic factors. Yet, more research is needed to characterize relevant exposures in IPF patients through the development of standardized questionnaires, identification and validation of circulating biomarkers, and investigations by industrial hygienists [107]. Moreover, further research should explore how gene–environment interactions affects disease development and progression. Such research has the potential to reduce the incidence of IPF through the identification of environmental causes of lung injury in genetically susceptible individuals.

## Figures and Tables

**Figure 1 ijms-24-16481-f001:**
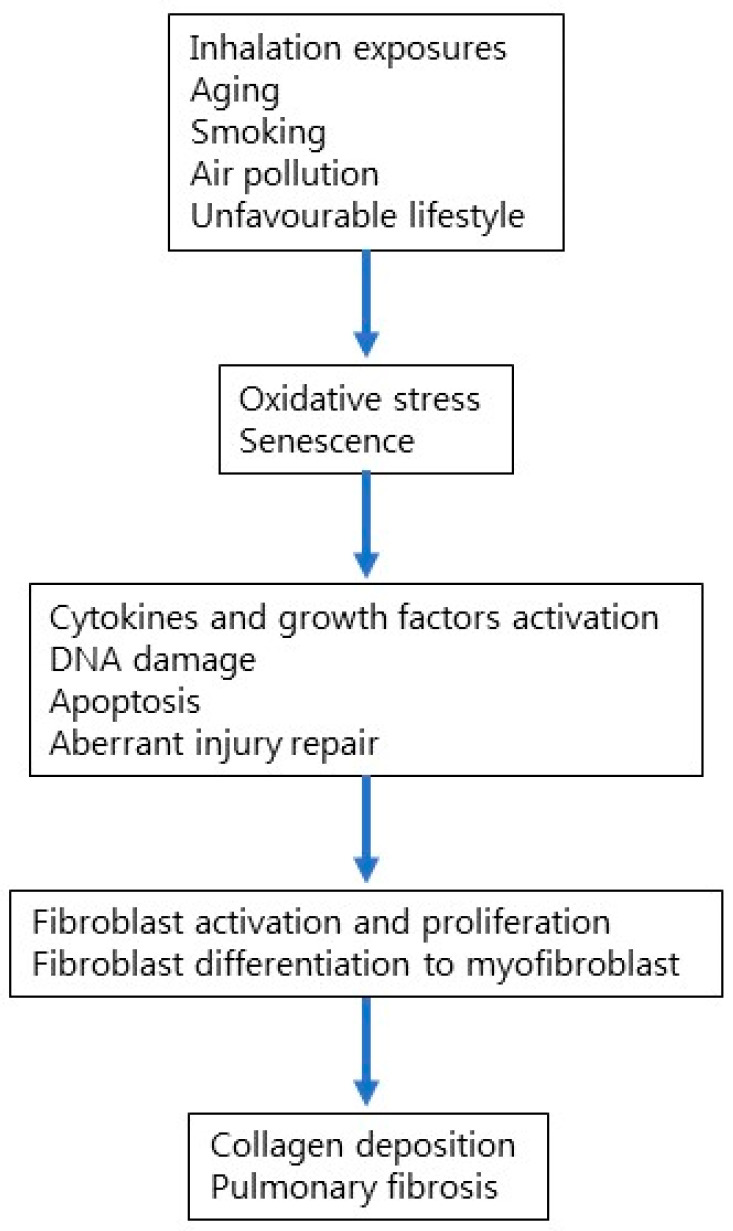
Cellular and molecular mechanisms involved in IPF pathogenesis.

**Figure 2 ijms-24-16481-f002:**
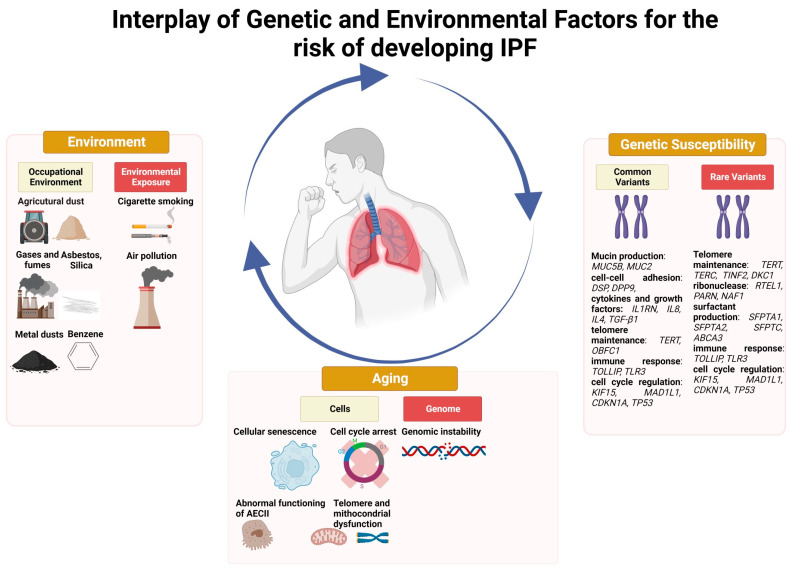
Interplay of genetic and environmental factors for the risk of developing IPF.

**Table 1 ijms-24-16481-t001:** Non-comprehensive list of known occupational etiologies of interstitial lung disease.

Exposure	Example Occupations	Interstitial Lung Disease
Aluminum dust	Welders, machinist aluminum products	Aluminum dust pneumoconiosis
Asbestos	Construction, shipyard worker, mechanic, insulator	Asbestosis
Beryllium	Nuclear weapon production, aerospace industry, ceramics	Chronic beryllium disease (indistinguishable from sarcoidosis)
Coal mine dust	Miner	Coal worker pneumoconiosis (black lung)
Indium	Semiconductor industry	Pulmonary alveolar proteinosis
Iron dust	Electric-arc and oxyacetylene welders	Pulmonary siderosis
Organic dust/animal antigens	Agricultural worker, poultry worker	Hypersensitivity pneumonitis
Silica dust	Mining, stone fabrication, sand blasting	Simple silicosis and progressive massive fibrosis
Talc	Talc miners and millers	Talc pneumoconiosis
Textile fiber dust	Clothing industry, Hemp, sisal, cotton production	Byssinosis
Tungsten-carbide	Machinist, dental technician	Hard metal lung disease (giant-cell interstitial pneumonitis)

## Data Availability

Not applicable.

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
