# Peer review of "Environmental Causes of Idiopathic Pulmonary Fibrosis"

_ijms, 2023, doi:10.3390/ijms242216481_

Round 1
Reviewer 1 Report
Comments and Suggestions for Authors
The submitted review entitled “Environmental causes of idiopathic pulmonary fibrosis” by Sheiphali Gandhi and colleagues addresses the different reasons to develop an IPF. The authors address molecular, environmental and genetic background mainly based on larges studies of patients. The manuscript is well-structured with mostly clear conclusion. However, there are some major points I would like to address.
Major:
First of all fibrosis is a result of chronic or sustained inflammation. This part is mi9ssing in the introduction. In line with this argument the role of the different environmental compounds toxins and even genetically change should be seen and discussed. Also shortly the pro-inflammatory mediators, which are later discussed in the different paragraphs.
The paragraph about aging (2.2) the main focus are epithelial type 2 cells. Here also other cell have to be discussed f.e. changes in macrophages (M1 vsM2), which are important in senescence.
In paragraph 2.4 IPF and gender is in focus. The authors mention the role of x-chromosome related genes. There some examples should be mentioned.
In Table 1 different exposures are mentioned. For fibers , asbestos and textile fibers are mentioned. What about other fibers types f.e. carbon nanotubes?
Paragraph 5 about terrorism and also Military exposures is somehow of context. The attack on WTC is just inhalation of dust in different compositions. The term Terrorism should not be used here. The last section about military exposures, does not add important information, as mechanisms are unknown.
Minor:
In paragraph 2.3 racial differences are discussed. Please use the term racial with caution!
Comments on the Quality of English Language-
Author Response
Please, see the attachment.

Reviewer 2 Report
Comments and Suggestions for Authors
The autors explane Idiopathic pulmonary fibrosis (IPF) as the most common and severe of the idiopathic interstitial pneumonias, as a chronic and relentlessly progressive disease, which occurs mostly in middle-aged and elderly males. Also, they focused on the first responders in the World Trade Center attacks and military exposure; and present an overview of the environmental and occupational causes of IPF and its pathogenesis.
The maniuscript is written in the standard English and stailish structured well (just some minor cheking is need).
Minor remarks-
1) a short chapter - oxidative disorders directly associated with IPF; The oxidative stress is very specific for IPF activation, and immuno re-activation.
2)Will be bettre if all of different parts will be accompanied by a scheme, for more visiability.
3)Table 1 is boring- to be reviced.
4)The conclusion - to be reviced.
5)Some of uced references are not according to the journal's requirements.
Comments on the Quality of English LanguageMinor editing of English language required
Author Response
Please, see the attachment.

Round 2
Reviewer 2 Report
Comments and Suggestions for Authors
-